# ADVERSARIAL OBJECT HALLUCINATION ATTACKS IN VIDEO-LANGUAGE MODELS VIA INTERMEDIATE FEATURE ALIGNMENT

## ABSTRACT

Video Large Language Models (Vid-LLMs) have rapidly advanced video under-standing, yet their robustness against semantic adversarial manipulation, espe-cially object hallucination, remains largely unexplored. We introduce Adversarial Object Hallucination (AOH), a novel attack that compels Vid-LLMs to "see" non-existent objects in videos by injecting visually imperceptible perturbations. Un-like prior attacks limited to inputs or outputs of videos, AOH directly manipulates intermediate connector features, aligning them with representations from a target video to induce controllable hallucinations. To systematically assess this threat, we curate a benchmark of 535 clean/target video pairs with high-quality VQA annotations. Extensive experiments show that AOH poses a severe threat to state-of-the-art Vid-LLMs, achieving highly effective attacks with alarming *cross-scale transferability*: adversarial examples optimized on smaller models transfer even more strongly to larger counterparts of the same architecture, amplifying attack impact while reducing adversarial cost. Further analyses reveal that perturbations encode semantic object contours, while Grad-CAM highlights their covert influ-ence. These findings expose a severe and previously overlooked vulnerability in Vid-LLMs, raising urgent concerns about their secure deployment and providing a foundation for future adversarial research in video-language modeling.

## 1 INTRODUCTION

In recent years, with the rapid advancements in deep learning, multimodal large models, especially Video Large Language Models (Vid-LLMs), have demonstrated unprecedented capabilities in un-derstanding and reasoning about video content. By integrating powerful vision encoders with large language models, these models can perform semantic analysis of complex video events, answer open-ended questions, and generate detailed descriptions, showcasing immense potential in critical domains such as autonomous driving, intelligent surveillance, and content creation. Prominent ex-amples include models like Video-ChatGPT (Maaz et al., 2024), VideoLLaMA (Zhang et al., 2023; Cheng et al., 2024; Zhang et al., 2025a), and InternVL (Chen et al., 2023).

However, the powerful capabilities of these models are accompanied by growing concerns regard-ing their security and robustness. In the image domain, adversarial attacks have become a mature research area, where attackers can induce models to make erroneous classifications, detections, or interpretations by adding subtle, often imperceptible, perturbations to images (Zhao et al., 2023; Hu et al., 2025; Wang et al., 2024; Zhang et al., 2025b). Despite this, the adversarial robustness of Vid-LLMs, particularly against semantic adversarial attacks that induce models to "see" specific, non-existent objects (i.e., object hallucination), remains a largely unexplored frontier in the dynamic and high-dimensional video domain. The temporal coherence and high dimensionality of video data pose unique challenges for generating adversarial examples, making existing image attack methods difficult to directly extend. This under-explored vulnerability, if exploited maliciously, could lead to severe consequences in critical applications MacLeod et al. (2017), such as misidentifying obstacles in autonomous driving systems or generating false alarms in security surveillance.

To bridge this critical gap, we propose Adversarial Object Hallucination (AOH), a novel adversarial attack. AOH aims to mislead Vid-LLMs into erroneously perceiving and reporting specific, non-

existent target objects within a video by introducing visually subtle perturbations to clean videos. Unlike prior attacks that directly manipulate the model's final output, our method uniquely focuses on manipulating the intermediate "Connector" feature representations within Vid-LLMs. We achieve this by aligning the intermediate features of an adversarial video with features derived from a pre-defined target video containing the desired object, thereby implanting the target object's semantic information deep within the model's internal representations.

To address the challenge of a lack of dedicated benchmark datasets for such attack tasks, we meticulously curate a comprehensive multi-source dataset. This dataset comprises 535 pairs of clean/target videos, rigorously annotated with Visual Question Answering (VQA) pairs, providing a robust foundation for comprehensively evaluating the vulnerability of Vid-LLMs to adversarial object hallucination attacks.

Concentrating on adversarial visual inputs, our work is distinct from previous adversarial attacks in two key aspects:

- **Attack Objective:** Our AOH attack aims to induce the *fabrication of non-existent semantic content* (i.e., object hallucination) within Vid-LLMs' understanding, leading to erroneous perceptions of objects that are not truly present. This fundamentally differs from traditional adversarial objectives like misclassification of existing objects, triggering harmful text outputs, or bypassing safety alignments. We focus on injecting a specific, fabricated visual semantic concept.
- **Attack Mechanism and Efficiency:** We achieve this through a intermediate feature alignment strategy, directly manipulating the high-level multimodal representations within the Vid-LLM's "Connector". Crucially, our findings reveal an alarming *cross-scale transferability*: adversarial examples generated for smaller models not only transfer successfully but often achieve *superior attack performance* on larger, more complex Vid-LLMs of the same architecture. This implies a potentially more efficient and lower-cost attack paradigm, as adversaries could target smaller, more accessible models to compromise larger, deployed systems.

In summary, our work not only uncovers a severe and insidious vulnerability in the semantic adversarial robustness of Vid-LLMs but also provides a pioneering framework for evaluating and understanding this vulnerability. Our findings emphasize the urgent need for developing Vid-LLMs with more robust internal representations and lay a foundational framework for future research in adversarial machine learning and multimodal AI safety. Our main contributions are as follows:

- We systematically investigate adversarial object hallucination in Vid-LLMs, revealing these models' susceptibility to semantic adversarial attacks in the video domain.
- We propose AOH, a novel attack method, which manipulates the intermediate "Connector" feature representations of Vid-LLMs to generate visually subtle adversarial video perturbations that efficiently induce precise object hallucinations.
- We construct and release the comprehensive benchmark dataset for video object hallucination attacks. This dataset includes 535 high-quality clean/target video pairs with rigorous VQA annotations, and due to its unique construction, it also holds potential value for related video editing tasks such as object removal.
- We conduct in-depth analyses of AOH's stealthiness. Using explainability tools like GradCAM, we confirm that the models' attention remains focused on naturally occurring regions in the video, rather than aberrantly on the hallucinated regions, when subjected to AOH attacks.

## 2 RELATED WORK

### 2.1 VIDEO LARGE LANGUAGE MODELS

The rapid advancements in Large Language Models (LLMs) such as GPT (Brown et al., 2020) and LLaMA (Touvron et al., 2023) have revolutionized natural language processing. This success has naturally extended to multimodal domains, leading to the development of Vision-Language Models (VLMs) that integrate visual information with text, exemplified by LLaVA (Liu et al., 2023) and MiniGPT-4 (Zhu et al., 2024). Building upon VLMs, Video LLMs (Vid-LLMs) further incorporate temporal dynamics to understand and reason about video content. These models typically consist

of a visual encoder (e.g., based on ViT or CLIP), a connector module to project visual features into the language model's embedding space, and a large language model (LLM) for conversational understanding and response generation. Prominent Vid-LLMs include Video-ChatGPT (Maaz et al., 2024), VideoLLaMA (Zhang et al., 2023; Cheng et al., 2024; Zhang et al., 2025a), InternVL (Chen et al., 2023), and LLaVA-OneVison (Li et al., 2025). These models excel in tasks such as video question answering (VQA) and video captioning, demonstrating sophisticated spatio-temporal reasoning capabilities. Current research primarily focuses on enhancing their performance, efficiency, and generalization across diverse video tasks.

## 2.2 Adversarial Robustness of Large Models

Adversarial attacks aim to mislead machine learning models by introducing subtle, often imperceptible, perturbations to their inputs. This field originated with attacks against image classification models, demonstrating vulnerabilities through techniques like FGSM (Goodfellow et al., 2015) and PGD (Madry et al., 2018). Extensive research has since focused on both developing novel attacks and designing robust defenses in computer vision.

The concept of adversarial robustness has also been extended to other modalities. In Language Models (LLMs), attacks typically involve textual perturbations such as typos, paraphrasing, or prompt engineering to induce factual errors, sentiment shifts, or jailbreaking behaviors (Zou et al., 2023; Yi et al., 2024; Jin et al., 2024). The focus here is often on semantic shifts and maintaining fluency while causing misinterpretation.

With the rise of multimodal models, robustness research has naturally expanded. Initial efforts investigated adversarial attacks against Vision-Language Models (VLMs), primarily in image-text scenarios (Zhou et al., 2024). These attacks aim to deceive VLMs in tasks like image captioning or visual question answering by perturbing images or text inputs, or both simultaneously. For instance, some work explores targeted attacks to make VLMs return predefined responses or misinterpret image content (Zhao et al., 2023; Hu et al., 2025; Wang et al., 2024; Zhang et al., 2025b). While these studies highlight vulnerabilities in multimodal understanding, they predominantly focus on static image-text inputs or broader unimodal attacks. Critically, although recent works have begun to address specific attacks on video-based models (Li et al., 2024; Huang et al., 2025), the adversarial robustness of Vid-LLMs, particularly concerning semantic manipulations such as inducing object hallucination within video sequences, remains a nascent and significantly under-explored area. Our work fills this gap by specifically targeting Vid-LLMs with semantic object hallucination attacks.

## 3 Dataset Construction

The task of adversarial object hallucination in Vid-LLMs necessitates a specialized dataset comprising clean video, target video, and corresponding ground-truth information such as object masks and VQA pairs. However, such a comprehensive benchmark is currently unavailable. To address this critical data scarcity and enable robust evaluation, we meticulously curate a multi-source benchmark dataset. This section details our data collection strategies, human annotation processes, and VQA generation methodology.

Our ideal approach to construct video pairs involves professionally adding realistic entities into clean videos to obtain (clean video, target video, target mask) triplets. While this offers high fidelity, it is technically demanding and highly time-consuming. To maximize data acquisition efficiently, we adopt a multi-pronged strategy: (1) leveraging existing public datasets that contain original and manipulated videos; (2) judiciously employing professional video editing for specific scenarios; and (3) ingeniously utilizing a "reverse thinking" approach by treating videos with existing objects as "target videos" and applying object removal techniques to generate corresponding "clean videos." This last strategy significantly expands our data pool, provided the removal quality is visually convincing.

## 3.1 Data Sources

We integrate and refine videos from four distinct sources to build our comprehensive dataset:

- **HQVI Dataset (Cho et al., 2025)**: This dataset provides high-quality, realistic Video Inpainting (VI) benchmark videos, synthesized by compositing objects from VideoMatte240K (Lin et al.,

2021) onto real-world Pexels videos using fine alpha mattes. HQVI offers multiple resolutions; considering that most video models compress resolution, we selected the 480P version, which suffices for most experimental model requirements. This dataset supplies 10 videos.

- **ROVI Dataset (Wu et al., 2024)**: ROVI is a pioneering dataset for object removal, containing triplets of original videos, removal expressions, and inpainted videos. For our "reverse thinking" approach, we treat the original videos as "target videos" and the inpainted videos as "clean videos." To ensure the highest visual realism for our "clean" videos (i.e., successfully removed objects), we introduced a rigorous human evaluation process. Three human annotators independently scored each inpainted video on a 5-point scale (with 5 being the best) across four dimensions: 1) Spatial Coherence / Visual Realism, 2) Temporal Consistency, 3) Artifact Severity, and 4) Overall Removal Quality. We aggregated scores from the three annotators and filtered for high-quality videos with a total score of 14 or 15. Due to challenges in providing adequate entity descriptions or persistent unnatural artifacts in some videos, we ultimately selected 468 videos for further annotation from the original 2,967 A2D-Sentences videos and 2,683 Refer-YouTube-VOS videos.

- **Video-Sham Dataset (Mittal et al., 2023)**: VideoSham is a video manipulation dataset featuring diverse, context-rich, human-centric manipulated videos created by professional video editors. We specifically selected videos from the "Adding an entity" and "Removing an entity" tasks. From "Adding an entity", we obtained 45 videos where the manipulated version serves as the "target video" and the original as the "clean video". For the "Removing an entity" task, similar to ROVI, we applied the same rigorous human filtering process to ensure high quality for the "clean videos", yielding 22 videos. In total, 67 videos were sourced from Video-Sham.

- **Custom Driving Dataset:** Recognizing the scarcity of driving scenarios in existing datasets—a crucial domain due to its implications for autonomous driving safety—we curated a bespoke dataset. This was constructed using real objects (e.g., vehicles, pedestrians, road signs) from the BDD100K dataset (Yu et al., 2020) within realistic driving scenes, employing alpha compositing via DaVinci Resolve video editing software, a process akin to HQVI. This dataset explicitly includes target masks. We generated 10 videos for this specific context.

In total, our benchmark dataset comprises **535 clean/target video pairs**, providing a robust foundation for evaluating adversarial object hallucination.

## 3.2 VQA Annotation

Accurate and unambiguous Visual Question Answering (VQA) pairs are essential for evaluating object hallucination attacks. We designed a multi-stage annotation pipeline leveraging both large language models (LLMs) and human expertise:

1. **Initial Information Acquisition:** For all target videos, we first gather essential auxiliary information: video descriptions (pre-generated by VideoLLaMA3-7B and refined by human annotators), target entity masks (obtained using Segment Anything 2 (Ravi et al., 2025)), and target entity text descriptions (manually added). For the ROVI dataset, which inherently provides these data, we directly utilized its existing annotations.

2. **VQA Pre-generation:** Leveraging advanced LLMs, specifically GPT-4o, Claude-3.5-Sonnet, and DeepSeek-V3, we pre-generate VQA pairs. This process utilizes the target video's description and the target entity's information to formulate pertinent questions regarding the presence and characteristics of the target object.

3. **VQA Refinement and Validation:** To minimize potential linguistic ambiguity or factual errors in the pre-generated VQA, we employ an iterative refinement process. The VQA pairs are tested against sampled frames (e.g., 4 frames) from the target video using the web-based ChatGPT. Based on the model's feedback and human review, the VQA pairs are repeatedly optimized until they are clear, precise, and accurately reflect the presence or absence of the target object.

## 4 ATTACK METHOD

This section formally introduces Adversarial Object Hallucination (AOH), our proposed white-box attack against Video Large Language Models. We begin by defining the threat model and then detail

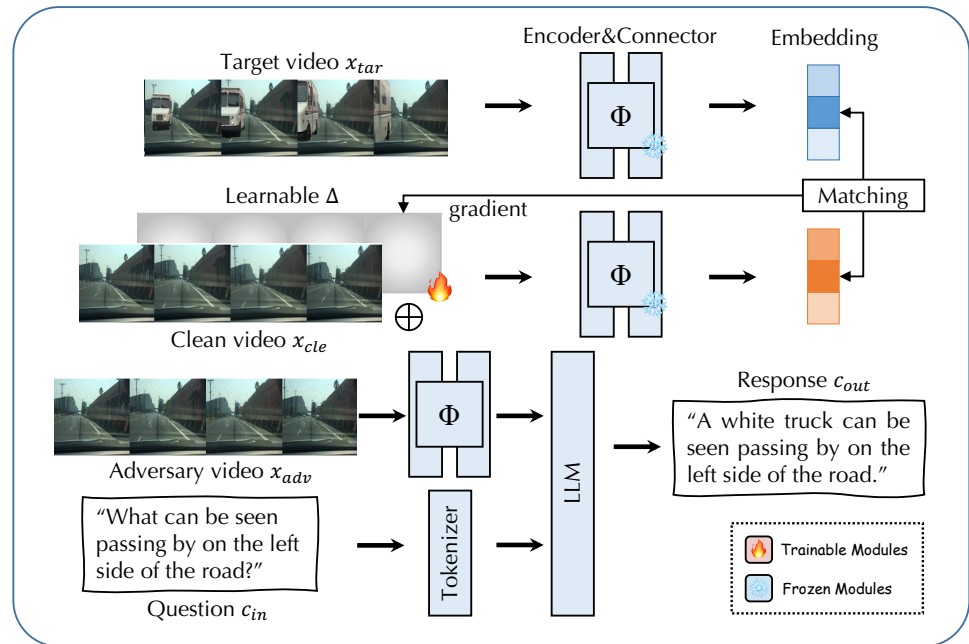

Figure 1: Pipelines of our attacking strategies.

our objective function, which leverages intermediate feature alignment to induce object hallucination with visually subtle perturbations.

## 4.1 THREAT MODEL

We denote a Vid-LLM as $F(x_v; c_{in}) \mapsto c_{out}$, where $x_v$ is the input video, $c_{in}$ is the input text (e.g., a question), and $c_{out}$ is the generated output text (e.g., an answer). The typical architecture of a Vid-LLM comprises a Vision Encoder and a Connector module, which together project the raw video input into the language model's embedding space. For simplicity, we denote this combined intermediate feature extraction process as $\Phi(x_v)$, where $\Phi(x_v)$ represents the output feature tensor from the Connector module for video $x_v$. Thus, the internal processing can be abstracted as $L(\Phi(x_v); c_{in}) \mapsto c_{out}$.

Our threat model specifies the adversarial conditions (Carlini et al., 2019) adapted for generating object hallucinations in Vid-LLMs:

1. **Adversary Knowledge:** The adversary is assumed to have *white-box access* to the victim Vid-LLM, $F$. This includes full knowledge of its architecture and weights, particularly those of the Vision Encoder and the Connector module (i.e., access to $\Phi(\cdot)$).

2. **Adversary Goals:** The adversary's primary goal is *targeted object hallucination*. Specifically, given a clean video $x_{cle}$ and a target object, the adversary aims to generate an adversarial video $x_{adv}$ such that the Vid-LLM $F$ perceives the target object as present in $x_{adv}$, even though it is absent in $x_{cle}$. This perception is evidenced by the model's VQA responses on meticulously designed benchmark.

3. **Adversary Capabilities:** The adversary can manipulate the input video $x_v$ to generate $x_{adv}$ by adding perturbations. The most critical constraint is imposed by the $\ell_p$ budget, ensuring the perturbations are *visually subtle*. We adopt the commonly used $\ell_\infty$ norm, such that $\|x_{cle} - x_{adv}\|_\infty \leq \epsilon$, where pixel values are in $[0, 255]$. Additionally, the adversary is *forbidden to manipulate the input text $c_{in}$*.

**Remark.** Our work investigates a challenging and realistic white-box threat model where the adversary leverages detailed model knowledge to induce specific semantic hallucinations with subtle perturbations in the high-dimensional video domain. Critically, while our attack is designed under white-box assumptions, our subsequent experiments reveal a remarkable **cross-scale transferability**

of these adversarial examples, indicating a broader and more concerning vulnerability in Vid-LLMs beyond the direct white-box setting.

## 4.2 INTERMEDIATE FEATURE ALIGNMENT

Figure 1 show the pipeline of our attacking strategies. Our AOH attack is designed to implant the semantic presence of a target object by aligning the intermediate feature representations of the adversarial video with those derived from a target video containing the desired object. Given a clean video $x_{cle}$ and a target video $x_{tar}$ (which contains the object we wish to hallucinate in $x_{cle}$), our objective is to find an adversarial video $x_{adv}$ such that:

$$x_{adv} = \underset{\|x_{cle}-x_{adv}\|_\infty \leq \epsilon}{\arg\max} \quad \text{cos\_sim}(\Phi(x_{adv}), \Phi(x_{tar})) \tag{1}$$

Here, $\text{cos\_sim}(\cdot, \cdot)$ represents the cosine similarity between the intermediate feature tensors from the $\Phi$ module for the adversarial video and the target video. By maximizing this similarity, we encourage the Vid-LLM's internal representation for $x_{adv}$ to embed the semantic information of the object present in $x_{tar}$, without directly manipulating the final text output layer. This approach subtly guides the model's perception at an earlier, feature-level stage, making the attack more robust and covert.

The constrained optimization problem in Equation (1) is solved iteratively using Projected Gradient Descent (PGD) (Madry et al., 2018). Specifically, for each iteration $t$:

$$x_{adv}^{(t+1)} = \text{Clip}_{x_{cle},\epsilon} \left[ x_{adv}^{(t)} + \alpha \cdot \text{sign} \left( \nabla_{x_{adv}^{(t)}} \text{cos\_sim}(\Phi(x_{adv}^{(t)}), \Phi(x_{tar})) \right) \right] \tag{2}$$

where $\alpha$ is the step size, and $\text{Clip}_{x_{cle},\epsilon}(\cdot)$ projects the perturbed video back into the $\epsilon$-neighborhood of the clean video $x_{cle}$ after each step, ensuring the $\ell_\infty$ perturbation budget is maintained. This iterative process allows us to subtly inject the target object's semantics into the video's intermediate representations, leading to the desired object hallucination by the Vid-LLM.

## 5 EXPERIMENTS

### 5.1 EXPERIMENTAL SETUP

All experiments were conducted on computing resources equipped with a NVIDIA 4090 GPU. Our experimental pipeline involves generating adversarial video samples using AOH, then evaluating the target Vid-LLMs' responses to both clean and adversarial videos using a meticulously designed VQA-based assessment framework.

**Target Video-Language Models** To provide a broad assessment of Vid-LLM vulnerabilities, we evaluate AOH against a diverse set of prominent open-source Vid-LLMs, selected for their varying architectures and scales. For consistency and resource efficiency, we used their default parameter settings but uniformly adjusted the sampled frame rate to 8 frames per video. Additionally, in line with prior work (Zhang et al., 2023; Cheng et al., 2024; Zhang et al., 2025a), the temperature setting for response generation is set to 0 during evaluation to ensure deterministic outputs:

- **Video-ChatGPT (Maaz et al., 2024)**: A 7B parameter model known for its strong video understanding capabilities.
- **VideoLLaMA3 (Zhang et al., 2025a)**: We experimented with both the 2B and 7B parameter versions of this model. A crucial consideration for VideoLLaMA3 is its dynamic resolution handling, which can lead to significant memory consumption for high-resolution videos [1]. To balance model accuracy and memory usage, we adaptively scaled its input resolution, restricting it to not exceed 480P (854x480) while maintaining aspect ratio. This adjustment ensures its maximum input size still generally exceeds that of other models in our experiments.
- **InternVL 2.5 (Chen et al., 2023)**: This model family offers a range of scales, and we tested the 1B, 2B, 4B, and 8B parameter variants.
- **LLaVA-OneVison (Li et al., 2025)**: We evaluated the 0.5B and 7B parameter versions of LLaVA-OneVison, which serve as key models for analyzing cross-scale transferability.

---

[1]https://github.com/DAMO-NLP-SG/VideoLLaMA3/issues/82

**Evaluation Metrics** Following established practices in evaluating VQA performance for multi-modal models (Maaz et al., 2024), we employ two primary metrics to quantify the success of object hallucination attacks and the overall video understanding capabilities of Vid-LLMs:

- **Accuracy:** This metric measures the binary correctness of a model's "yes/no" answer to a question. For clean videos, a high Acc indicates accurate understanding of the absence of the target object. For adversarial videos, a low Acc (i.e., incorrect "no" answer) or a high Acc (i.e., incorrect "yes" answer that aligns with hallucination) indicates successful deception.

- **ChatGPT-3.5 Assisted Scoring (0-5 Scale):** Inspired by previous work (Maaz et al., 2024), we utilize GPT-3.5 to provide a more nuanced qualitative assessment of the model's generated responses. For each VQA pair, GPT-3.5 evaluates the model's answer against the ground-truth answer (for clean videos) or the target answer (for adversarial videos), assigning a score from 0 (completely incorrect/irrelevant) to 5 (excellent, semantically aligned, and detailed). High scores on clean videos reflect strong baseline understanding. For adversarial videos, high scores indicate that the model not only perceived the hallucinated object but also provided semantically rich and accurate details about it, signifying a more profound attack success.

**Attack Implementation Details** Our AOH attack aims to induce object hallucination with visually subtle perturbations. We compare our method against several baselines:

- **Clean (None):** The original, unperturbed video. This serves as the baseline for ideal model performance, where Vid-LLMs are expected to correctly identify the absence of the target object.

- **Random Noise (Random):** Adversarial videos are generated by adding random Gaussian noise to the clean video. The noise is scaled such that its maximum perturbation matches our $\ell_\infty$ budget, i.e., with a standard deviation derived from $\epsilon^2$, mirroring settings in prior (Li et al., 2024). This baseline assesses whether generic noise can induce hallucination.

- **Random Noise with Mask Assistance (Random&Mask):** Similar to the Random Noise baseline, but noise is applied only within the spatial and temporal regions defined by the target object's mask. This explores whether localizing random perturbations to the target area enhances hallucination.

- **Our Attack (AOH):** We implement AOH using Projected Gradient Descent (PGD) to optimize the intermediate feature alignment objective. We perform 300 optimization steps. The perturbation budget is set to $\epsilon = 8$ under an $\ell_\infty$ constraint, i.e., $\|x_{cle} - x_{adv}\|_\infty \leq 8$. This is a widely adopted setting in adversarial literature (Madry et al., 2018; Zhao et al., 2023) to ensure that adversarial perturbations remain visually subtle, given that pixel values are normalized to $[0, 255]$.

**Dataset Usage** For all experiments, we employ our carefully curated multi-source benchmark dataset, which contains 535 clean/target video pairs. Each clean video $x_{cle}$ and its corresponding target video $x_{tar}$ is annotated with VQA pairs (e.g., "Is there a car passing the intersection?") and their ground-truth answers: CA (the correct answer when the queried object is absent) and TA (the correct answer when the queried object is present). In the evaluation, when the queried object is absent, models are expected to generate answers consistent with CA. Conversely, when the queried object is supposed to appear, a hallucination is considered successful if the model's response matches TA, indicating that it perceives the non-existent object.

## 5.2 ATTACK PERFORMANCE

We evaluate the overall effectiveness of our proposed AOH attack against various Vid-LLMs, comparing it with baseline attack methods (Random Noise, Random Noise with Mask Assistance) and the model's performance on clean videos (None baseline). Table 1 summarizes the Acc and ChatGPT-3.5 Score across all tested models.

Across all evaluated Vid-LLMs, our AOH attack consistently achieves significantly higher object hallucination rates (Acc) and more semantically rich hallucinated descriptions (Score) compared to all baselines. The "None" baseline, representing the model's false positive rate on clean videos, shows relatively low Acc (ranging from 0.141 to 0.348) and Score (from 1.153 to 2.148). Random noise baselines ("Rand." and "Rand.&Mask") offer only marginal improvements over "None", indicating that generic or localized random perturbations are largely ineffective for inducing targeted se-

Table 1: Main experimental results of AOH attack on various Vid-LLMs. "Clean" and "Target" represent the model's baseline performance on unperturbed clean and target videos, respectively. "None", "Random", "Random&Mask", and "Ours" denote the performance under no attack, random noise, masked random noise, and our AOH attack. Higher Acc and Score under "Ours" indicate greater attack success. "Evaluation" assesses the model's basic capability: "Clean" uses $x_{cle}$ as input with CA as the answer, while "Target" uses $x_{tar}$ as input with TA as the answer.

| Vid model | Metric | Evaluation | | Attacking method | | | | Other info. | |
|---|---|---|---|---|---|---|---|---|---|
| | | Clean | Target | None | Rand. | R.&M. | Ours | # Param. | Res. |
| LLaVA-Onevision 0.5B | Acc | 0.449 | 0.593 | 0.157 | 0.159 | 0.200 | **0.607** | 894M | 384 |
| | Score | 1.863 | 3.198 | 1.279 | 1.319 | 1.514 | **3.258** | | |
| InternVL2.5 1B | Acc | 0.350 | 0.580 | 0.209 | 0.218 | 0.223 | **0.413** | 938M | 448 |
| | Score | 1.659 | 3.236 | 1.477 | 1.490 | 1.526 | **2.438** | | |
| VideoLLaMA3 2B | Acc | 0.480 | 0.142 | 0.141 | 0.146 | 0.164 | **0.495** | 1.96B | Dyn. |
| | Score | 1.986 | 1.173 | 1.153 | 1.182 | 1.220 | **2.695** | | |
| InternVL2.5 2B | Acc | 0.461 | 0.614 | 0.187 | 0.204 | 0.211 | **0.506** | 2.21B | 448 |
| | Score | 1.881 | 3.321 | 1.389 | 1.423 | 1.503 | **2.820** | | |
| InternVL2.5 4B | Acc | 0.384 | 0.631 | 0.245 | 0.245 | 0.263 | **0.395** | 3.71B | 448 |
| | Score | 1.692 | 3.411 | 1.596 | 1.618 | 1.681 | **2.277** | | |
| Video-ChatGPT 7B | Acc | 0.123 | 0.582 | 0.348 | 0.346 | 0.362 | **0.402** | ~8B | 224 |
| | Score | 0.838 | 3.169 | 2.148 | 2.117 | 2.180 | **2.348** | | |
| LLaVA-Onevision 7B | Acc | 0.434 | 0.596 | 0.142 | 0.153 | 0.227 | **0.386** | 8.03B | 384 |
| | Score | 1.850 | 3.249 | 1.314 | 1.319 | 1.611 | **2.359** | | |
| VideoLLaMA3 7B | Acc | 0.438 | 0.166 | 0.168 | 0.175 | 0.180 | **0.586** | 8.04B | Dyn. |
| | Score | 1.906 | 1.323 | 1.337 | 1.323 | 1.377 | **3.209** | | |
| InternVL2.5 8B | Acc | 0.381 | 0.640 | 0.227 | 0.214 | 0.240 | **0.465** | 8.08B | 448 |
| | Score | 1.692 | 3.423 | 1.553 | 1.559 | 1.634 | **2.609** | | |

mantic hallucinations. In stark contrast, AOH significantly boosts Acc (e.g., LLaVA-OneVison 0.5B from 0.157 to 0.607, VideoLLaMA3 7B from 0.168 to 0.586) and Score (e.g., LLaVA-OneVison 0.5B from 1.279 to 3.258, VideoLLaMA3 7B from 1.337 to 3.209), demonstrating its superior capability in implanting specific object semantics.

Notably, for models like LLaVA-OneVison 0.5B, AOH's Acc (0.607) even surpasses the model's performance on the actual target videos (Target Acc: 0.593), suggesting that AOH can make the model perceive non-existent objects more reliably than it detects truly present ones. Similarly, for VideoLLaMA3 7B, AOH's Score (3.209) is exceptionally close to the model's Score on target videos (3.249 from InternVL2.5 8B), showcasing the high fidelity of the hallucinated object's description. VideoLLaMA3 2B and 7B exhibit surprisingly low "Target" Acc and Score, indicating a struggle to accurately identify or describe the object even when explicitly present. However, AOH still dramatically increases their Acc and Score, forcing them to hallucinate effectively, which underscores AOH's ability to manipulate underlying feature representations regardless of the model's inherent detection robustness.

## 5.3 EXPERIMENTAL ANALYSIS

**Cross-Scale Transferability** A critical finding of our study is the alarming cross-model transferability of AOH adversarial examples. We investigate this phenomenon by generating adversarial videos using LLaVA-OneVison-0.5B and LLaVA-OneVison-7B, and then evaluating their performance when applied to the other model. Results are presented in Table 2. The most striking observation is the transferability from the smaller LLaVA-OneVison-0.5B to its larger counterpart, LLaVA-OneVison-7B. Adversarial samples crafted for the 0.5B model achieve an Acc of 0.533 and a Score of 3.018 when attacking the 7B model. This performance is remarkably higher than the 7B model's own-generated adversarial samples (Acc 0.386, Score 2.359). This surprising result highlights a dangerous possibility: attackers can design adversarial samples against smaller, computationally less expensive models, and these samples not only transfer to larger, more complex

Question $c_{in}$: What, if any, is moving through the sky above the winding river?
CA: Nothing is moving through the sky, just a clear blue sky over the landscape
TA: A bald eagle is soaring through the sky

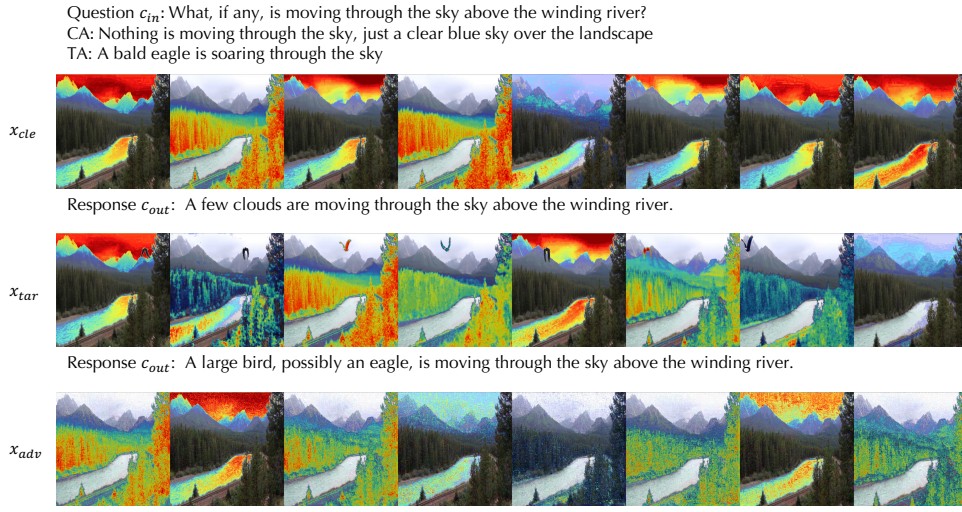

$x_{cle}$

Response $c_{out}$: A few clouds are moving through the sky above the winding river.

$x_{tar}$

Response $c_{out}$: A large bird, possibly an eagle, is moving through the sky above the winding river.

$x_{adv}$

Response $c_{out}$: A large bird, possibly an eagle, is soaring through the sky above the winding river.

Figure 2: GradCAM visualizations on a clean video (top), its target video (middle), and AOH adversarial counterpart (bottom).

systems but often achieve superior attack efficacy at a fraction of the cost. Conversely, adversarial samples generated for the 7B model show reduced, though still present, attack performance when transferred to the 0.5B model (Acc 0.312, Score 1.998), suggesting an asymmetric transferability where attacks from smaller to larger models are particularly potent. This discovery exposes a profound and cost-effective vulnerability in Vid-LLMs, demanding urgent attention.

**GradCAM for Attack Mechanism Explanation** To shed light on how Vid-LLMs process AOH-perturbed videos, we extend the traditional GradCAM technique to video-text tasks, providing visual explanations of model attention. Using LLaVA-OneVison-7B as an example, our GradCAM visualizations reveal that even when the model successfully hallucinates an object, its attention is not unnaturally focused on the regions where the target object is supposedly introduced. Instead, the model's attention predominantly remains on naturally existing objects and salient areas within the video. This indicates that our AOH attack is highly covert; it manipulates the model's internal representations without forcing an overt shift in its visual attention towards the hallucinated entity, making the attack difficult to detect through traditional interpretability methods. Visual examples of GradCAM on clean and adversarial videos are presented in Figure 2.

Table 2: Cross-model transferability results for AOH attacks between LLaVA-OneVison 0.5B (LLaVA-OV 0.5B) and 7B models (LLaVA-OV 7B). "Before" refers to the attack performance on the original target model (e.g., 0.5B generated for 0.5B). "After" refers to the attack performance when transferred to the other model (e.g., 0.5B generated then tested on 7B).

| Vid model | Before | | After | |
|---|---|---|---|---|
| | Acc | Score | Acc | Score |
| LLaVA-OV 0.5B | **0.607** | **3.258** | 0.312 | 1.998 |
| LLaVA-OV 7B | 0.386 | 2.359 | **0.533** | **3.018** |

## 6 CONCLUSION

In this paper, we introduced Adversarial Object Hallucination (AOH), a novel white-box attack revealing semantic vulnerabilities in Video Large Language Models (Vid-LLMs). Through intermediate feature alignment, AOH effectively induces targeted object hallucinations with visually subtle perturbations, significantly outperforming baselines. Critically, we uncovered an alarming **cross-scale transferability** where adversarial examples crafted for smaller models achieve superior efficacy on larger Vid-LLMs of the same architecture, presenting a potent and cost-effective threat. Our analysis further confirmed AOH's covert nature, manipulating models via semantically structured perturbations without obvious shifts in visual attention. This work highlights a severe vulnerability in Vid-LLMs, underscoring the urgent need for robust internal representations and providing a foundational framework for future adversarial research in multimodal AI safety.

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
