# OpenReview forum: "Adversarial Object Hallucination Attacks in Video-Language Models via Intermediate Feature Alignment"
_ICLR.cc/2026/Conference — ICLR 2026 Conference Withdrawn Submission_

### Official Review · Reviewer_N33z · 2025-10-25

**Soundness:** 3
**Presentation:** 1
**Contribution:** 2
**Rating:** 2
**Confidence:** 4

**Summary:**

This work investigates adversarial attacks on the video domain. Focusing on creating object hallucination, an area that is currently not fully investigated. It does so by aligning specific intermediate features. It also constructs a new benchmarking dataset.

**Strengths:**

1. This work investigates object hallucination, an area currently not fully explored in adversarial attacks.
2. The author collected and annotated a benchmarking dataset, which facilitates fair and handy comparison for future works

**Weaknesses:**

1. The novelty of the proposed method is limited. The general attack framework highly resembles the classical encoder-alignment methods in the VLM attack [1~5]. The only modification is moving the alignment from the encoder feature to the connector's feature.
      >Also, there is no ablation study for the proposed method. Even using a simple method, valuable insights can be gathered from ablation studies. The author chose to align the connector's feature. What happens if the alignment is altered to the vision encoder/text features? What happens if all three are aligned? A proper comparison of performance, along with a detailed analysis, can further enrich the method section and justify the simple yet effective approach.
2. How $\alpha$ is set and selected is unknown.
3. Lack of other cross-series transfer results. The author only reports LLaVA-OV 0.5B/7B transfer results. These two models are from the same family, possibly distilled from a single teacher model, and thus share similar behavior. Cross-series transfer can further convince the effectiveness of this method.
4. Presentation. Table 2 should use the row and column headers as the source/target model.  The current before/after on the row makes reading difficult.

I believe in the potential of the proposed method and the author's effort in collecting this dataset. But the authors spend too much time describing the dataset collection, with only a simple method proposed, making the full paper feel like a benchmarking & dataset track submission, which is not aligned with the current main area of the submission. Therefore, at this time, this submission is below the acceptance level.

**References**

[1] Liu D, Yang M, Qu X, et al. A survey of attacks on large vision-language models: Resources, advances, and future trends[J]. arXiv preprint arXiv:2407.07403, 2024.

[2] Zhao Y, Pang T, Du C, et al. On evaluating adversarial robustness of large vision-language models[C] Advances in Neural Information Processing Systems, 2023, 36: 54111-54138.

[3] Dong Y, Chen H, Chen J, et al. How robust is google's bard to adversarial image attacks?[J]. arXiv preprint arXiv:2309.11751, 2023.

[4] Yin Z, Ye M, Zhang T, et al. Vlattack: Multimodal adversarial attacks on vision-language tasks via pre-trained models[C] Advances in Neural Information Processing Systems, 2023, 36: 52936-52956.

[5] Li, Zhaoyi, et al. "A frustratingly simple yet highly effective attack baseline: Over 90% success rate against the strong black-box models of gpt-4.5/4o/o1." _arXiv preprint arXiv:2503.10635_ (2025).

**Questions:**

Please see the above weaknesses section.

**Details Of Ethics Concerns:**

The methodologies can be used to produce the harmful videos targeting the production system.

On the other hand, the method can facilitate robust training.

It should be the author's responsibility to clarify it. However, as an adversarial attack method, the original submission lacks an Ethics Statement section.

---

### Official Review · Reviewer_xvVX · 2025-10-29

**Soundness:** 3
**Presentation:** 2
**Contribution:** 2
**Rating:** 6
**Confidence:** 3

**Summary:**

This paper presents Adversarial Object Hallucination (AOH), a novel white-box attack that compels Video-Language Models (Vid-LLMs) to perceive and describe non-existent objects in videos via intermediate feature alignment. Rather than perturbing raw pixels or final outputs, AOH manipulates the Connector features of Vid-LLMs by maximizing cosine similarity between the adversarial and target video representations. The authors also curate a benchmark of 535 clean/target video pairs with VQA annotations to systematically evaluate object-hallucination robustness. Experiments across multiple Vid-LLMs (VideoLLaMA3, LLaVA-OneVision, InternVL 2.5, Video-ChatGPT) show AOH induces strong and transferable hallucinations, particularly from smaller to larger model scales, while maintaining imperceptible perturbations. Grad-CAM analyses indicate the attacks remain visually covert.

**Strengths:**

- This work introduces an intermediate feature alignment strategy that injects target semantics into Connector embeddings rather than logits, offering an interesting and underexplored direction for adversarial research in Vid-LLMs.
- Curates 535 clean/target video pairs by integrating HQVI, ROVI, Video-Sham, and BDD100K sources, with rigorous human filtering and LLM-verified VQA annotations. This benchmark enables quantitative evaluation of hallucination attacks, a contribution not seen in prior work.
- Demonstrates consistent attack success across eight Vid-LLMs (0.5B–8B parameters). Grad-CAM visualizations indicate that model attention remains focused on natural regions, suggesting stealthy internal manipulation by the proposed method.

**Weaknesses:**

- The study evaluates only under a white-box threat model. The black-box practicality and real-world feasibility remain untested, limiting external validity.
- Limited comparison with related works, for example, CAVALRY-V [1], and its baselines [2–3].
- Benchmarks focus solely on VQA; no experiments are conducted on open-ended captioning, reasoning, or safety-critical tasks where hallucinations could be particularly harmful.
- While effective, cosine-similarity-based feature alignment is well established. The key novelty lies in applying it to Connector features, but no theoretical analysis or empirical evidence is provided to justify why this layer choice is optimal.
- The paper identifies vulnerabilities but offers no mitigation strategies, defense baselines, or robustness analyses.

[1] Zhang, J., Hu, R., Guo, Q., & Lim, W. Y. B. (2025). CAVALRY-V: A Large-Scale Generator Framework for Adversarial Attacks on Video MLLMs. arXiv preprint arXiv:2507.00817.\
[2] Huang, H., Erfani, S. M., Li, Y., Ma, X., & Bailey, J. (2025). X-Transfer attacks: Towards super transferable adversarial attacks on CLIP. In Proceedings of the 42nd International Conference on Machine Learning (Vol. 267, pp. 25204–25234).\
[3] Zhang, J., Ye, J., Ma, X., Li, Y., Yang, Y., Chen, Y., ... & Yeung, D. Y. (2025). AnyAttack: Towards Large-scale Self-supervised Adversarial Attacks on Vision-language Models. In Proceedings of the Computer Vision and Pattern Recognition Conference (pp. 19900-19909).

**Questions:**

- Could the authors extend evaluations to black-box or transfer-based settings to demonstrate broader applicability?
- Could the author provide some comparison with the suggested related works? It would be good to discuss them in the related works sections as well.
- Why is selecting Connector features particularly effective for inducing object hallucination? Some theoretical justification or ablation study would strengthen the argument.

---

### Official Review · Reviewer_UR3H · 2025-10-30

**Soundness:** 3
**Presentation:** 4
**Contribution:** 2
**Rating:** 2
**Confidence:** 3

**Summary:**

The submission is motivated by the looming threat of adversarial image patterns for mission-critical applications using vision language models, or specifically, video large language models. The authors focus specifically on adversarial object hallucination (AOH) since it may affect downstream applications, leveraging known feature priors from  intermediate feature representations of the multi-modal fusion blocks in Vid-LLMs. The authors first collect a multi-source benchmark dataset that aims to represent (clean, target, target mask) triplets without having to add target objects manually. The authors propose to leverage existing datasets, including ROVI, Video-Sham, and HQVI, alongside a custom driving scenario dataset. For ROVI, the authors propose to use reverse inpainting to remove subjects from the video samples, effectively making them "clean" videos. These are later manually evaluated to filter for visual artifacts and realism. The target videos are fed into an annotation pipeline to generate text descriptions of the targets, create VQA pairs with existing LLMs, and validated against a handful of frames with ChatGPT. The threat model can be summarized as white box with bounded $l_\inf$-norm such that the adversary may observe the target Vid-LLM. The authors propose to use PGD to optimize a video noise mask, such that the perturbed video semantic features are closer to those of the target video in the Vid-LLM connector. The attack is tested on a variety of Vid-LLM models from different architectures and model families, including InternVL and LLaVA, alongside ablations to check performance for random noise, mask-assisted random-noise (only noise in the target spatial dimensions). The authors analyze the attack success rate for each, check the cross-scale transferability, and check the high-level receptive field with GradCAM.

**Strengths:**

- The writing quality is excellent and conveys the technical points well. The notation appears to be free of major issues and is consistent with familiar convention in the field. Figures and tables are rendered properly and there were no issues viewing them at standard zoom levels.
- The submission investigates the important concern of adversarial samples in Video-Language models, particularly the ability for those patterns to force hallucination via the high-level semantic alignment of a noise mask with a target video.
- The authors highlight an interesting artifact of the AOH attack, which is the ability to generate successful attacks using smaller Vid-LLMs for larger models.
- Through GradCAM visualizations, it is shown that the Vid-LLM attention may not necessarily be focused on the region of forced hallucination. Hence it may be that the semantic features (textures or shapes) of the target are embedded in other spatial regions of the frames.

**Weaknesses:**

- For some results it is difficult to check the statistical significance. In Table 1 for example, I suspect the false negatives/positives are highest for the smaller models, so the proposed attack accuracy may be within margin of error from the clean accuracy. The results for 7B and 8B models are more encouraging, although without an explicit measure such as standard deviation, it is difficult to make claims on models, for example InternVL2.5 which had 0.381 clean vs. 0.465 AOH. The same issue applies for the results in Section 5.3.
- The dataset size is described in terms of amount of clean/target video pairs, but it can be difficult to gauge the scale since video datasets may have diverse clip lengths and resolutions. On L189 the authors should clarify the total runtime (minutes/hours) of the dataset and summarize their resolution, it appears to always be 480p but it wasn't clear if this was only HQVI. For VQA annotation, it should be clarified how much text was generated in number of tokens.
- Overall the submission offers some interesting first looks at a few different aspects of video hallucination (success rate, receptive field, and transferability) but each aspect is not allocated much depth to have definitive takeaways. Specifically:
    - The attack success rates show AOH has potential, but the analysis currently lacks statistical rigor. It would also help if the authors offered preliminary results for transferability to black-box systems, since that is a natural consequence of the use-cases described in the introduction, and preliminary results offer a first look that may not be available yet in the broader field.
    - There is evidence of cross-scale transferability using smaller models, but ultimately it is still not well understood why the transferability occurs, so it is difficult for any action to be taken. The authors should investigate this more deeply since it seems the most interesting thread for a contribution.
    - The Vid-LLM high-level receptive field does not necessarily attend to the hallucinated object, but it is also not well explored beyond the surface level evidence. I would expect those high-level semantic patterns to be embedded throughout the spatial dimensions since the loss gradient does not force those concepts to appear in the target region. Some deeper investigation of this behavior would strengthen the contribution. If the authors choose to stay with the white-box setting, then going deeper in this direction would probably offer more takeaways to the community.

**Questions:**

- I would be interested to see the accuracy and scoring results split among dataset type, since the current results seem to take the aggregate of all video types (non-driving video + driving video).
- Since the authors propose a new video dataset, they should carefully consider the distribution rights associated with the videos and clarify if they will be shared or not. If the dataset is distributed to the wider community, it should be properly annotated to clarify that some annotations were machine-generated.

---

### Official Review · Reviewer_vB5F · 2025-10-30

**Soundness:** 3
**Presentation:** 3
**Contribution:** 3
**Rating:** 2
**Confidence:** 4

**Summary:**

This paper investigates adversarial object hallucination in Video Large Language Models. It proposes Adversarial Object Hallucination, an attack that manipulates Vid-LLM's intermediate "Connector " features to align with target video features, using visually imperceptible perturbations.

**Strengths:**

- Well written.
- The proposed method is interesting.

**Weaknesses:**

- This paper claims adversarial examples from small models (e.g., LLaVA-OneVison 0.5B) attack large models (e.g., LLaVA-OneVison 7B) more effectively, but provides no causal explanation.
- The transferability experiment only uses models of the same architecture (e.g., LLaVA-OneVison 0.5B $\rightarrow$ 7B).  No tests on cross-architecture transfer (e.g., VideoLLaMA 2B $\rightarrow$ InternVL 8B) are conducted.
- AOH is claimed to induce "controllable hallucinations ", but the VQA evaluation is limited to binary "yes/no " questions (e.g., "Is there a car? "). No tests of fine-grained hallucinations (e.g., "Is there a red Tesla? ") are conducted.
- Can fine-tuning Vid-LLMs on the AOH benchmark reduce hallucination?  Or can modifying the Connector to add noise to aligned features mitigate the attack?
- This paper seems to have migrated PGD in terms of methodology, but there are too few baseline experiments for comparison. Why not compare it with various variants of PGD (such as MI-FGSM [1], NI-FGSM [2], PI-FGSM [3], VMI-FGSM [4], etc.)?
- This paper only considers the gradient-based method to update $x_{adv}$. Then why wasn't the optimization-based method (such as C&W [5]) taken into account?

[1] Dong, Yinpeng, et al. "Boosting adversarial attacks with momentum." Proceedings of the IEEE conference on computer vision and pattern recognition. 2018.

[2] Lin, Jiadong, et al. "Nesterov accelerated gradient and scale invariance for adversarial attacks." arXiv preprint arXiv:1908.06281 (2019).

[3] Gao, Lianli, et al. "Patch-wise attack for fooling deep neural network." European Conference on Computer Vision. Cham: Springer International Publishing, 2020.

[4] Wang, Xiaosen, and Kun He. "Enhancing the transferability of adversarial attacks through variance tuning." Proceedings of the IEEE/CVF conference on computer vision and pattern recognition. 2021.

[5] Carlini, Nicholas, and David Wagner. "Towards evaluating the robustness of neural networks." 2017 ieee symposium on security and privacy (sp). Ieee, 2017.

**Questions:**

- Please see "Weaknesses".

---

### Note · Authors · 2025-11-12

I have read and agree with the venue's withdrawal policy on behalf of myself and my co-authors.